# Neonatal Resuscitation with an Intact Cord: Current and Ongoing Trials

**DOI:** 10.3390/children6040060

**Published:** 2019-04-22

**Authors:** Anup C. Katheria

**Affiliations:** 1Sharp Mary Birch Hospital for Women & Newborns, San Diego, CA 92123, USA; anup.katheria@sharp.com; 2School of Medicine, Loma Linda University, Loma Linda, CA 92350, USA

**Keywords:** delayed cord clamping, resuscitation, newborn, premature infants

## Abstract

Premature and full-term infants are at high risk of morbidities such as intraventricular hemorrhage or hypoxic-ischemic encephalopathy. The sickest infants at birth are the most likely to die and or develop intraventricular hemorrhage. Delayed cord clamping has been shown to reduce these morbidities, but is currently not provided to those infants that need immediate resuscitation. This review will discuss recently published and ongoing or planned clinical trials involving neonatal resuscitation while the newborn is still attached to the umbilical cord. We will discuss the implications on neonatal management and delivery room care should this method become standard practice. We will review previous and ongoing trials that provided respiratory support compared to no support. Lastly, we will discuss the implications of implementing routine resuscitation support outside of a research setting.

## 1. Introduction

There is a large body of evidence that has now demonstrated that delayed cord clamping (DCC) has benefits for term and preterm infants. [1,2,3]. These benefits include reductions in morbidities such as intraventricular hemorrhage (IVH) and mortality in preterm infants and improved developmental scores at 4 years of age in term infants. One important group that has not been included is the nonbreathing, non-vigorous infant. It may be important to determine whether an infant has started to breathe before the umbilical cord is clamped. This has been suggested in a number of small studies where infants who breathed had less blood retained in the placenta and higher hematocrits [4,5]. Preterm infants also take longer to establish breathing than term newborns, placing greater emphasis on the determination of breathing before the cord is clamped [6,7]. One epidemiologic study suggested that term newborns that take longer to breathe after their cord is clamped have a higher rate of admission to the NICU and increased mortality [8]. Another observational study of preterm infants demonstrated higher rates of IVH and bronchopulmonary dysplasia (BPD) in infants that were not breathing during delayed cord clamping [9].

Animal work suggests that positive pressure ventilation in anesthetized preterm lambs during DCC eliminated fluctuations in cardiac output and cerebral blood flow compared to early cord clamping [10]. One hypothesis is that in response to the lungs inflating, the placenta provides the additional blood supply needed to fill the pulmonary vascular bed. The venous return from the lungs then enters the left atrium and provides the needed preload for left ventricular output and cerebral perfusion. The result is increased blood volume and stable perfusion, particularly to the brain [11]. This is critical particularly in preterm infants that have limited cerebral autoregulation (Figure 1).

The challenge with translating physiology into evidence lies with the ability to include infants that require resuscitation into randomized controlled trials of delayed cord clamping. Various trials have had significant protocol violations, including the largest trial (Australian Placental Transfusion Study, APTS, *n* = 1500) in which over 25 percent of infants that were randomized to delayed cord clamping actually received immediate cord clamping [2]. The major reason cited was due to concerns for the infant. Unfortunately, infants that need greater resuscitation are more likely to have greater morbidity including IVH and death [12]. Despite a number of smaller trials and animal data suggesting DCC could reduce IVH, the APTS trial showed no difference in IVH. Facilitating a placental transfusion in this population could have greater benefits than only the included vigorous preterm infants that have been included in previous trials. Respiratory support, in the form of positive pressure ventilation or continuous positive airway pressure (CPAP) could be given to all infants undergoing DCC or only to those who fail to establish spontaneous respirations. Alternatively, more invasive strategies such as endotracheal intubation may be provided.

There are several completed and ongoing trials investigating ventilation with an intact umbilical cord. Since there is significant heterogeneity with respect to the population (term versus preterm) and feasibility studies, we will discuss these in their respective sections.

## 2. Feasibility Studies in Preterm Infants

There are several recently completed feasibility trials involving resuscitation with an intact cord. The first pilot study by Winter et al. sought to determine whether ventilatory support with continuous positive airway pressure or positive pressure ventilation was feasible during 90 s of DCC in infants at 24 to 32 weeks gestation [13]. A total of 29 infants were enrolled and all completed the protocol without any adverse outcomes.

Blank et al. recently completed a feasibility trial using a colorimetric CO_2_ detector to manage when cord clamping was performed [14]. The trial enrolled 44 infants and the median time to cord clamping was 150 s and 138 s in vigorous and non-vigorous infants, respectively.

Knol et al. completed a feasibility study to demonstrate whether infants <35 weeks could be placed on a specialized bed equipped with respiratory function monitoring long enough to achieve physiological targets before having cord clamping. These targets were set as a heart rate >100 bpm, spontaneous breathing on continuous positive airway pressure with tidal volumes >4 mL/kg, and SpO_2_ ≥ 25th percentile and fraction of inspired oxygen (FiO_2_) <0.4. They were able to achieve these targets in 33/37 infants. The median time to cord clamping was 4 min and 23 s after birth [15].

Presti et al. completed a pilot feasibility study comparing 3 min delay with resuscitation when needed to intact umbilical cord milking four times. Infants with delayed cord clamping had a higher 5 min Apgar, but a lower admission temperature [16].

## 3. Completed Randomized Controlled Trials

To date, there are three randomized controlled trials involving resuscitation with an intact cord (Table 1). The first trial by Duley et al. randomized infants to immediate cord clamping (<20 s) or at least 2 min of delayed cord clamping with resuscitation when needed [17]. The median gestational age was 28 weeks. There was no difference in any outcomes, but a trend for decreased mortality with delayed cord clamping emerged. However, only 59 percent of the infants randomized to delayed cord clamping reached the 2-minute time point.

Our group performed a randomized trial and compared initiation of ventilation immediately after birth compared to no ventilation at all for 60 s with an intact cord in preterm infants <32 weeks (*n* = 150) [18]. We also conducted a second randomized clinical trial in term babies at risk for resuscitation (*n* = 60), comparing resuscitation with an intact cord to usual care [19]. Neither study found any clinical differences. However, in our preterm trial, over 90 percent of infants began breathing before 60 s with gentle stimulation in both groups. If spontaneous ventilation in humans mimics the improved and stable hemodynamics found in the animal model, this may explain why positive pressure ventilation had no value.

## 4. Ongoing Randomized Controlled Trials

There are three ongoing trials involving resuscitation with an intact cord (Table 2). The first trial, VentFirst (NCT02742454, proposed *n* = 940) aims to determine whether ventilation during delayed cord clamping of up to 120 s reduces the incidence of intraventricular hemorrhage compared to delayed cord clamping with no respiratory support for 30 s in non-breathing infants or 60 s in spontaneously breathing infants of 23–28 + 6 weeks gestation at delivery. The trial currently has about *n* = 200 infants (personal communication) with an expected completion of June 2023.

Ashish et al. have recently completed a single center trial in Nepal (Nep-Cord 3 trial, *n* = 231, NCT02727517) in depressed term neonates to compare delayed cord clamping of greater than 180 s with resuscitation with early cord clamping (<60 s). The primary hypothesis is that delayed cord clamping improves oxygen saturation, heart rate, and Apgar scores in the first 10 min of life compared to early cord clamping.

Blank et al. are conducting a similar study (Baby DUCC, Australian Trial Registry #12618000621213) in non-vigorous infants (*n* = 120) greater than or equal to 32 weeks gestational age. Infants will receive delayed cord clamping with resuscitation until at least 1 minute after the colorimetric CO2 detector has indicated gas exchange or at 5 min of life they will receive immediate cord clamping. The primary outcome in this trial will be the heart rate at 60 and 120 s. The trial has enrolled 30 infants and is expected to be completed in 2021 (personal communication).

Knol et al. are conducting a multicenter trial of time-based clamping (*n* = 660, ABC2 NCT03808051) 30–60 s after delivery depending on the condition of the infant) compared to physiological-based cord clamping in infants less than 30 weeks gestation. The outcome is a composite outcome of intact survival (survival without grade 2 IVH or higher, or necrotizing enterocolitis).

Nevill et al. are studying preterm infants that are not breathing well by 15 s of life and randomizing them to breathing assistance or just stimulation until 60 s of life. The primary outcome is the need for a blood transfusion.

Lastly, Presti et al. are proposing a multicenter trial (PCI-Trial, NCT02671305) comparing intact umbilical cord milking to delayed cord clamping for 3 min with assistance when needed in premature infants <32 weeks. Their primary outcome is a composite outcome of severe IVH, BPD, and death with a sample size of 202 infants.

## 5. Challenges with Resuscitation with an Intact Cord

There have been two surveys to report parental and provider assessments of performing resuscitation with an intact cord. Thomas et al. evaluated term and preterm newborns that received resuscitation with an intact cord [20]. The majority of clinicians in their study (86%) rated the resuscitation trolley as the “same”, “better”, or “much better” than conventional equipment. In a second survey involving term infants done by our group, we found that only 41% of neonatal providers and 32% of maternal providers found the process of resuscitation with an intact cord as positive or strongly positive [21]. One-third felt that the cord was too short for access and they had poor access to the baby. Twenty-five percent also felt that they had difficulty accessing equipment. On the other hand, for both surveys, the majority of parents felt that the experience of bedside resuscitation was positive. This was probably because they were able to visualize and or touch their newborn immediately after delivery and throughout the resuscitation.

In practice, many of these concerns are still a challenge. With many of our studies on resuscitation with an intact cord currently facing challenges with maintaining a sterile field (there is not a commercially available sterile T-piece), providing adequate temperature (a chemical mattress had to be used on top of the resuscitation bed), and only one provider had access to equipment. A patient could not have oxygen or ventilator pressures easily adjusted. In addition, there was not an available display for monitoring the patient viewable to the provider. A new purpose-built resuscitation table has been developed to attempt to overcome some of these issues [15].

## 6. Conclusions

Neonatal resuscitation with an intact cord is an exciting possibility to ensure every newborn receives a placental transfusion. Several important clinical trials in both term and preterm infants will help answer the question as to its clinical benefit. There are a number of logistical issues that will need to be improved before this becomes standard practice, but it holds tremendous promise.

## Figures and Tables

**Figure 1 children-06-00060-f001:**
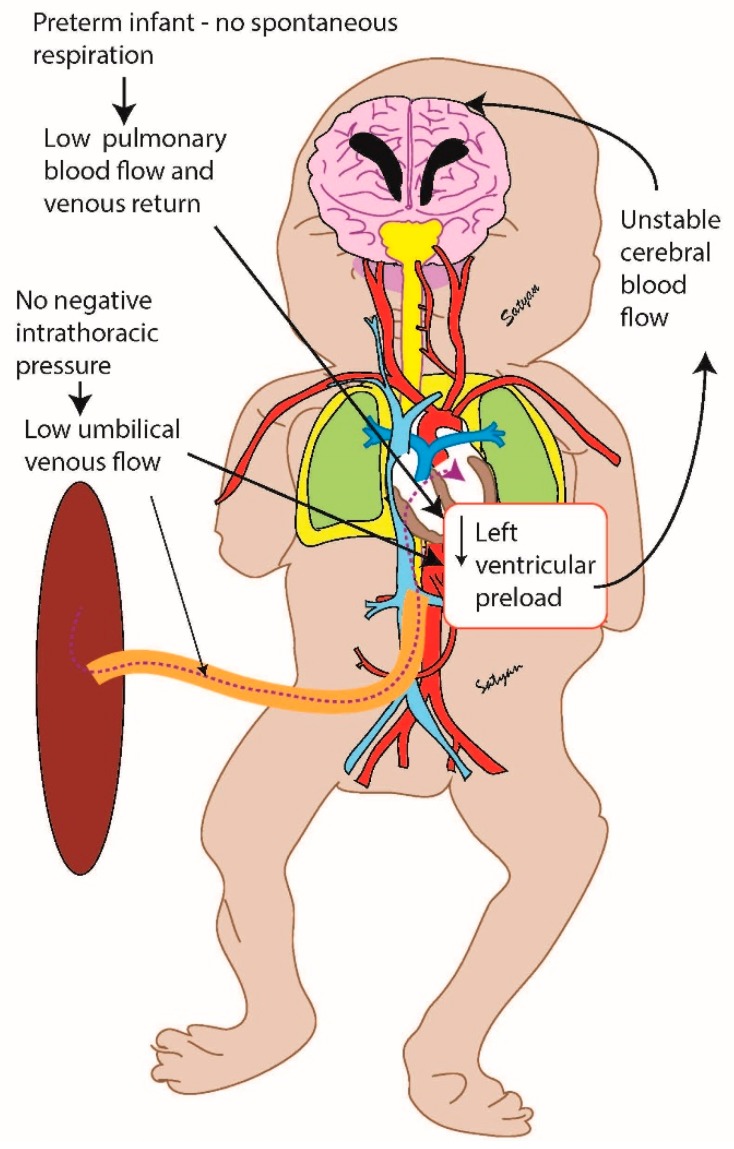
Preterm infant without spontaneous respirations.

**Table 1 children-06-00060-t001:** Completed trials of neonatal resuscitation with an intact cord.

Study	*n*	GA (weeks)	Intervention	Time of Cord Clamping, Control Arm	Time of Cord Clamping Intervention Arm	Clinical Outcome
Duley et al., 2017 [17]	137	23–31	Resuscitation if needed	20 s	120 s	No difference
Katheria et al., 2016 [18]	154	23–31	CPAP and PPV if apneic	60 s	60 s	No difference
Katheria et al., 2018 [19]	60	37–42	Resuscitation if needed	60 s	3–5 min	No Difference

**Table 2 children-06-00060-t002:** Ongoing or planned trials of neonatal resuscitation .

Study	Proposed N	GA (weeks)	Intervention	Time of Cord Clamping, Control Arm	Time of Cord Clamping Intervention Arm	Primary Outcome
VentFirst	940	23–28	CPAP 30–120 s	30–60 s	120 s	IVH
Nep-Cord 3	231	37–41	Resuscitation if needed	<60 s	180 s	SpO2, HR, and Apgar scores in first 10 min
Baby DUCC	120	32–41	Resuscitation if needed	Immediate cord clamping (duration not specified)	Until 1 minute after CO_2_ detector change or 5 min	Heart Rate at 60 and 120 s
ABC2	660	24–31	Resuscitation if needed	30–60 s	Until stable (approx. 4 min)	Intact Survival (survival without grade 2 ivh or nec)
Nevill and Meyers	120	23–31	Start CPAP and or PPV at 15 s until 60 s	60 s	60 s	Need for blood transfusion
PCI-Trial	202	23–31	Resuscitation if needed	30–60 s	3 min	Composite outcome of severe IVH, BPD, and death

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
