# Peer review of "Neonatal Resuscitation with an Intact Cord: Current and Ongoing Trials"

_children, 2019, doi:10.3390/children6040060_

Reviewer 1 Report

Thank you for this excellent review.

I agree that neonatal resuscitation with an in tact cord is an exciting possibility. The review succinctly reviews the available evidence and highlights the ongoing trials in this area.

I would only like to make two suggestions:

A table may enhance the readers experience. I suggest this as each trial has different inclusion criteria (eg gestation) and interventions (eg time for DCC etc). Although the information is provided as coherently as possible, a Table may further help summarize past/present research in this area.

Practical issues are a major concern for centres wishing to take part in research trials/ provide neonatal resuscitation with in tact cords. Major concerns are infant temperature; maintaining a sterile field during cesarean sections (eg plastic bag); access to equipment etc. I think this article could benefit with suggestions on how to overcome these obstacles and whether in practice they are an issue in the author's experience? 

Author Response

Thank you for your comments. I have added a short paragraph about the challenges of resuscitation with an intact cord and two additional tables for completed and ongoing trials. Thank you again.

Reviewer 2 Report

This short review about Neonatal Resuscitation with an intact umbilical cord is from a well-known author who has performed randomized controlled studies in both preterm and term infants.  This approach is novel as delayed cord clamping facilitates placental transfusion and stabilizes the circulation and hemodynamics post resuscitation. In particular, it is now well established that delayed cord clamping reduces the incidence of IVH in premature babies. However, the process of resuscitating a newborn with intact umbilical cord is challenging as described by the author’s own experience from their surveys. Nevertheless, the ongoing trials will establish the benefits and long-term outcomes of this novel approach. It does appear that resuscitations with an intact umbilical cord is feasible and facilitates placental transfusion. This review is well written and covers all the trials from feasibility, completed and ongoing trials.

Author Response

Thank you for your kind comments.

Reviewer 3 Report

Thank you for the opportunity to appraise Dr. Katheria's review on neonatal resuscitation with an intact umbilical cord. Dr. Katheria is a leading physician scientist and clinical investigator in newborn resuscitation with an intact umbilical cord. A brief review of the physiologic benefits and the feasibility of conducting resuscitation with an intact cord are presented, followed by an overview of current clinical trials. 

While the focus is on resuscitation with an intact umbilical, this reviewer thinks that adding a brief discussion on cord milking will be of interest. 

Author Response

Thank you for your comments and review. While I would love to add a short section on cord milking. I think it would not do the ongoing authors much justice since I am very biased towards that method. There are also many methods (cut and intact cord milking) and even removing the placenta intact and resuscitation that could be described. I think trying to have an unbiased review of resuscitation with an intact cord is probably the best method moving forward.

However if you reply and disagree I will certainly consider it.